# Knowledge and Perception of Registered Nurses Regarding the Scope of Practice of Speech-Language Pathologists

**DOI:** 10.3390/ijerph181910534

**Published:** 2021-10-08

**Authors:** Sami Alhamidi, Majid Alshahwan, Regie Tumala

**Affiliations:** 1Postgraduate and Research Center, Department of Maternal and Child Health, College of Nursing, King Saud University, Riyadh 12372, Saudi Arabia; salhamidi@ksu.edu.sa; 2Speech and Hearing Program, Department of Rehabilitation, College of Applied Medical Sciences, King Saud University, Riyadh 12372, Saudi Arabia; mialshahwan@ksu.edu.sa; 3Medical-Surgical Nursing Department, College of Nursing, King Saud University, Riyadh 12372, Saudi Arabia

**Keywords:** interprofessional collaboration, knowledge, perception, registered nurse, Saudi Arabia, speech-language pathologist

## Abstract

*Objective:* To assess the knowledge and perception of registered nurses regarding the scope of practice of speech-language pathologists (SLP) in Saudi Arabia. *Background:* Interdisciplinary collaboration is vital between the SLP and nurses due to the complex care needed by patients with speech problems. *Methods:* A total of 294 registered nurses were surveyed utilizing the Knowledge and Perception of Speech-Language Pathologists questionnaire. Descriptive statistics and tests for differences and relationships were performed. *Results:* The findings revealed that nursing respondents had an overall good understanding of the value and scope of practice of the SLP. However, they frequently and wrongly responded to scenarios concerning patients who suffered from Alzheimer’s dementia, laryngeal cancer, traumatic brain injury, and stroke. *Conclusions:* It is vital for nurses to understand the value, role, and scope of practice of the SLP. Further educational nursing interventions and training are necessary for effective interprofessional collaboration and teamwork.

## 1. Introduction

A speech-language pathologist (SLP) is a member of the healthcare team who primarily performs the assessment, evaluation, and treatment of swallowing disorders [1]. Speech-language pathology therapy (SLPT) prevents and corrects language, speech, voice, and fluency problems in patients [2]. Thus, assessment and intervention for patients with speech problems are primarily the responsibility of the SLP [2,3,4]. Patients with these problems most commonly suffer from swallowing disorders resulting from cerebrovascular accidents or stroke attacks [1] or mild traumatic brain injury in children [5]. Every year in the United States of America (USA), around 600,000 patients with neurological impairments are affected by swallowing difficulties [6]. Fortunately, these patients can rely on a support system of, on average, 52.8 SLP for every 100,000 residents [7]. However, there are no known data regarding this ratio for Saudi Arabia [8].

Due to the complex care needed by patients with speech problems, a teamwork approach with interprofessional collaboration between the SLP and other healthcare workers, particularly nurses, is paramount [1]. According to the World Health Organization [9], the SLP must educate and collaborate with other healthcare professionals to enhance the understanding, appreciation, and management of patients with swallowing disorders. This effort will improve coordination between healthcare services, ensure appropriate referrals are made by specialists, enhance the patient’s health outcomes, and enact safer care for the patient [10]. Due to many collaborative efforts, the perceptions of other healthcare professionals including nurses play an important role in the appropriate and timely referral of patients with speech problems to SLP. A recent study revealed that the majority of Jordanian dentists (N = 191) generally believed in the important role of SLP in the healthcare team [11].

However, these collaborative efforts are challenged as several studies reported a lack of knowledge and poor understanding of the role and scope of practice of SLP. For example, a previous study using similar tool revealed poor knowledge regarding the role of SLP among allied healthcare professionals in an acute care hospital setting in the USA [12]. The findings were consistent with a previous study conducted in Canada showing that medicine, nursing, physiotherapy, and occupational therapy students reported a lack of understanding of the role of SLP [13]. In addition, a recent study that involved 290 American undergraduate male students towards the SLP field indicated a lack of awareness of the scope of SLP [14]. Another study in the USA found that caregivers struggled to understand their child’s language and literacy disorders, underscoring the need for SLP to provide a clear diagnostic and clinical service [15]. Similarly, public awareness of speech pathology among residents of the Central Queensland community in Australia was reported as low [16]. In Pakistan, a recent study participated in by 200 healthcare professionals including audiologists, dieticians, doctors, nurses, physiotherapists, and psychiatrists, reported low perception and unfamiliarity of the role of SLP, and poor communication and referral to SLP [17]. In particular, a previous study among South African nurses indicated a lack of knowledge and unfamiliarity with the role of SLP in dysphagia management [18]. Hence, there is evidence in the literature that the role and scope of the practice of SLP are poorly understood by the public and healthcare professionals including nurses as reported in many countries, suggesting the need for effective interprofessional collaboration.

In the USA and Canada, interdisciplinary collaborations have been established between the SLP and audiologists [3], occupational therapists, physical therapists [19], and nurses [1,20]. The fact that these collaborations are occurring is even evident in SLP and nursing, facilitating interdisciplinary competence and teamwork for new areas of knowledge and skills [4,10,21]. Interprofessional collaboration has been reported in a previous study in the USA to be an effective strategy to increase understanding of the roles of healthcare professionals through simulation involving nursing, nutrition, and SLP students [22]. In addition, some recent, related studies among the SLP, audiologists, and nurses have suggested these collaborations are happening in Saudi Arabia [8,23], which is imperative to ensure the highest quality and safest care is delivered to patients [21]. However, the literature suggests that there remains a paucity of published studies on nurses collaborating with the SLP [18,20]. This lack of research is particularly prevalent in Saudi Arabia, where SLPT is still a relatively new profession [8]. Moreover, there has only been one study in Saudi Arabia specifically focused on patients with dysphagia [23].

Thus, the present study was undertaken to assess the interdisciplinary knowledge and perceptions of registered nurses regarding the role and scope of practice of the SLP in a Saudi state university medical facility.

## 2. Methods

### 2.1. Study Design and Setting and Participants

This quantitative study used a correlational, cross-sectional design. A convenience sample of 294 registered nurses was recruited at King Saud University Medical City in Riyadh, Saudi Arabia. During the recruitment period, 320 questionnaires were distributed, 302 were returned, and 8 were excluded due to substantial missing data. Thus, the response rate was 94.38%.

The inclusion criteria were as follows: registered nurses with at least six months of current employment in the hospital who willingly and voluntarily participated and were available at the time the data were collected. Newly hired nurses under probationary status and nurses who were unwilling to participate were excluded.

### 2.2. Instrument

This study utilized the Knowledge and Perception of Speech-Language Pathologists questionnaire [12] with three parts. Permission to use the questionnaire was obtained from the copyright holder, Elizabeth King [email approval, 12 March 2019]. The tool had been used in previous studies [12,17], and was pilot-tested among 50 nurses in Saudi Arabia before the start of the study, which yielded alpha values > 0.70 for the two sub-scales: (1) value of SLPT and (2) scope of practice of SLP.

The first part had eight questions asking the nursing respondents about their work-related information. The second part of the questionnaire had four main questions about the respondents’ perceptions of the value of SLPT. The first question, which had six sub-questions, was about the importance of the SLP in delivering services to a patient who (a) had a brain injury or stroke, (b) was on a ventilator for several days, (c) was in a persistent vegetative state with tube feeding, (d) was having trouble swallowing food, (e) had acute vomiting due to a bowel obstruction, and (f) was receiving head/neck radiation. These sub-questions were answered using a four-point Likert scale from 1 (being not at all important) to 4 (being extremely important). The three remaining questions were concerned with the patient’s communicative abilities, patient’s swallowing abilities, and whether or not the SLPT services made a considerable contribution to patient care. These questions were answered with “Yes”, “No”, or “Uncertain”. The three questions were merely matters of opinion and there were no right or wrong answers. If the respondents responded with “Yes” on each question, they had a positive perception of SLP; whereas if they responded with “No” on the questions, they had a negative perception of SLP.

The third part of the questionnaire had eleven scenario questions about the respondents’ understanding of the scope of practice of the SLP. The respondents were asked to indicate all healthcare professionals who should be involved in the patients’ treatment, with choices including the SLP, dietician, nurse, occupational therapist, physical therapist, pharmacist, social worker, and others. To receive credit for a correct response, the respondents had to mark the SLP as one of the healthcare professionals who should be involved in the patient’s treatment for each of the following items: 1, 2, 4, 5, 6, 8, 9, and 11. The SLP should not be marked for the following foil items: 3, 7, and 10. It did not matter if other disciplines were or were not marked on this part of the questionnaire. SLP was the only discipline that determined whether the responses were correct or incorrect. The acceptable range of performance on the scale was 80% correct or higher.

### 2.3. Ethical Consideration and Data Collection

Ethical approval with reference no. E-19-4126 was obtained from the Institutional Review Board (IRB) of King Saud University. After obtaining ethical approval, the researchers were granted permission by the office of the nursing director to distribute the survey. Written informed consent was then obtained from each participant before administering the questionnaire. Afterwards, the questionnaires were handed to the participants ensuring that no signed consent form was linked to any completed survey. Adequate information was provided to the respondents during the recruitment phase. Moreover, the respondents were told about their rights and were informed that no incentives were given for participation. The respondents completed the survey in between 15 and 20 min. Data were collected between October 2019 and December 2019.

### 2.4. Statistical Analysis

Descriptive statistics were calculated for the work-related variables of the respondents (e.g., median, mean, and standard deviation). A Pearson correlation test was performed to establish the association between the work-related characteristics and the perception ratings and the scenario scores. The results between the groups of respondents were then compared. An independent *t*-test and one-way analysis of variance were performed to examine differences on the perceptions and understanding of the value of the SLPT. Significant findings were inferred if *p* < 0.05. Data were processed and analyzed using IBM SPSS Statistics for Windows v.23 (IBM Corp.: Armonk, NY, USA).

## 3. Results

### 3.1. Work-Related Characteristics of the Participants

The work-related variables of the participants are presented in Table 1. The mean years of experience of the respondents in the hospital were 7.65 (SD = 5.53), ranging from 0 to 37 years. The highest proportion of the respondents was in the emergency and out-patient department (17.0%), while the lowest proportion was in the psychiatric department (4.4%). The number of nurses working a day shift (77.2%) was higher than those working a night shift (22.8%). The majority of the nurses (240, 81.6%) had not communicated with or referred to the SLP in the last month, while 41 (13.9%) had done so 1–10 times, and 13 (4.4%) had done so more than 10 times. Similarly, the majority of the nurses (222, 75.5%) had not cared for a patient with the SLP in the past month, while 48 (16.3%) had cared for 1–10 patients, and 24 (8.2%) had cared for more than 10 patients. Of particular note, the highest proportion of nurses was unfamiliar with the role of the SLP in the hospital setting (62.6%), while only four (1.4%) were very familiar. The number of nurses who had not personally received the SLPT services or known anyone who had (87.1%) was higher than those who had (12.9%).

### 3.2. Participants’ Perception of the Value of Speech-Language Pathology

As can be seen in Table 2, most of the nurses perceived the SLP as important or extremely important for providing services to patients who have had a stroke or brain injury (92.1%), with the highest mean reported in this study (M = 3.58, SD = 0.74). Within the sample, the majority of the nurses perceived the SLP as important or extremely important for providing services to patients who have been on ventilators for several days (75.2%) and have had trouble swallowing food (2.4%). The highest proportion of nurses also perceived the SLP as important or extremely important for providing services to patients receiving head/neck radiation (64.7%). On the other hand, foil questions were set to a reverse scale. Low ratings were considered more appropriate than high ratings and thus were worthy of a greater amount of points. For these questions, the nurses did not perceive the SLP to be important or possibly important for patients in a persistent vegetative state who received tube feeding from nurses (63.3%), with a reported mean of 2.21 (SD = 1.14). This finding shows that the respondents gave an appropriate rating for this situation, indicating a very good perception of the value of SLPT. Another foil question regarding a patient suffering from acute vomiting due to a bowel obstruction (52.4%) was also appropriately perceived by the respondents by rating it as not at all important or possibly important, with a reported mean of 2.52 (SD = 1.15). This finding shows that the nurses had a good perception of the value of SLPT in this situation. Lastly, the majority of the respondents believed a patient’s communicative (87.1%) and swallowing (68%) abilities could change as a result of SLPT and believed that the SLPT services made a considerable contribution to patient care within a hospital setting (85.4%).

### 3.3. Differences and Associations between the Work-Related Characteristics and Perceptions of the Value of Speech-Language Pathology

Table 3 presents the results of the tests for differences and associations between the work-related variables and the nurses’ perceptions of the value of SLPT. The number of nurses who had not personally received the SLPT services or known anyone who had was found to have a significant difference to those who had (*t* = −2.45, *p* = 0.015). Of additional significance, the familiarity of the nurses with the role of the SLP in the hospital setting was significantly associated with their perceptions of the value of SLPT (*r* = 0.13, *p* = 0.032). Other work-related variables were not found to have significant differences and associations with the nurses’ perceptions of the value of SLPT.

### 3.4. Participants’ Understanding of the Speech-Language Pathologist’s Scope of Practice Based on Scenarios

The respondents’ understanding of the scope of practice of the SLP in given scenarios is presented in Table 4. The overall percentage (64.4%) of correct answers from the nurses was below the acceptable range of performance (80% correct or higher). However, the majority of the nurses performed acceptably when only four out of the eleven scenarios were included: Scenario 4 (84.0%), Scenario 6 (82.0%), Scenario 7 (82.0%), and Scenario 9 (81.3%). The patients in Scenarios 4 and 9 suffered a left hemisphere stroke, while the patient in Scenario 6 suffered lapses in memory and periods of disorientation that required cognitive therapy, which is within the scope of practice of the SLP. Although not counted as a missed question, Scenario 7 was also included on the survey as a foil and described a patient who suffered nausea and vomiting, which would not fall within the scope of practice of the SLP.

Despite these positive responses, most of the nurses answered incorrectly in seven scenarios, with the percentage of correct answers far below the acceptable range of performance in Scenario 1 (73.8%), Scenario 2 (57.8%), Scenario 3 (65.0%), Scenario 5 (45.2%), Scenario 8 (41.8%), Scenario 10 (18.4%), and Scenario 11 (77.6%). Of these, Scenarios 3 and 10 were included on the survey to serve as additional foils. The two patients in these scenarios were described as follows: (1) a patient with diabetes and a mild mental disability and (2) a patient having hallucinations due to the street drug commonly called Spice. Neither patient had conditions or circumstances that would fall within the scope of practice of the SLP. However, Scenarios 1, 2, 5, 8, and 11 were not foils. The patients in these scenarios suffered from a traumatic brain injury (Scenarios 1 and 5), a stroke (Scenario 2), Alzheimer’s dementia (Scenario 8), and laryngeal cancer (Scenario 11). In each situation, the patient would require cognitive therapy, which is within the scope of practice of the SLP.

### 3.5. Differences and Associations between the Work-Related Characteristics and Understanding of the Speech-Language Pathologist’s Scope of Practice

Table 5 presents the results of the tests for differences and associations between the work-related variables and the nurses’ understanding of the scope of practice of the SLP. The work-related variables were not found to have significant differences and associations with the nurses’ understanding of the scope of practice of the SLP.

## 4. Discussion

This study assessed the knowledge and perceptions of SLPT among staff nurses in Saudi Arabia. The study presented critical findings on nurses’ knowledge and perceptions of SLPT. As a result, these findings can help improve educational interventions and collaborative practices among nurses to ensure appropriate knowledge and a correct understanding of the scope of practice of the SLP is gained.

Based on the findings of the study, the nurses exhibited very good perceptions of the value of SLPT for providing services to patients who have had a stroke or brain injury, have been on a ventilator for several days, have had trouble swallowing food, and have had received head or neck radiation. Importantly, the nurses were correct in rating foil questions for providing services to patients who have had a persistent vegetative state with tube feeding and have had acute vomiting due to a bowel obstruction. The nurses indicated that it was very important for the SLP to provide services to these patients. The very good perception of the respondents regarding this finding means that the nurses perceived that the SLPT provided by the SLP could substantially improve a patient’s swallowing and communicative abilities. The findings are consistent with a recent study that involved 191 dentists in Jordan where the majority of the participants generally supported the importance of the role of SLP in the healthcare team [11]. However, several studies in the USA presented opposing results indicating poor knowledge and a lack of understanding of the role and value of SLP among allied healthcare professionals [12], caregivers of children with speech problems [15], and male students towards the SLP profession [14]. In addition, other contrary findings were also reported among allied healthcare students including nursing students in Canada [13], community residents in Australia [16], healthcare professionals including nurses in Pakistan, and among registered nurses in South Africa [18]. The work-related characteristics of the respondents in this study and those studies in other countries (e.g., Australia, Canada, Jordan, Pakistan, South Africa, and the USA) may explain the possible reasons for the conflicting views and perceptions of the role and value of SLP.

On the other hand, the nurses performed poorly when demonstrating their understanding of the scope of practice of the SLP. Only 64.4% of questions were answered correctly, which is lower than the acceptable range of performance of 80% correct or higher. This finding is alarming since it indicates a lack of awareness among the nurses about SLPT. Since the interprofessional practice between the SLP and nurses is essential [22,24], the nurses’ capability to collaborate is a critical component of the professional practice in order to deliver holistic and patient-centered care [1]. The most frequently missed questions involved patients who had suffered a traumatic brain injury, stroke, Alzheimer’s dementia, and laryngeal cancer and therefore needed cognitive therapy. The majority of the nurses answered that the patient should not be referred to the SLP, demonstrating their limited understanding of cognitive and language therapy, which is likely the result of their limited experience collaborating with the SLP. The scope of practice for the SLP embraces cognition therapy which includes the emerging practice areas of attention, executive functioning, memory, and problem solving [7,12]. Thus, the patients in these scenarios should receive the SLPT services. This finding is consistent with a previous study in Nova Scotia, Canada, which concluded that students in medicine, nursing, occupational therapy, and physiotherapy possessed inadequate knowledge regarding the role of the SLP in cognitive and social language therapy [13]. In addition, another Canadian study suggested that the amount of time spent by the SLP providing communication intervention for non-speaking adults in acute care settings was relatively minimal [20].

In contrast, the nurses who had not personally received the SLPT services or known someone who had were found to have significant differences from those who had. This finding shows that the respondents who had not personally received the SLPT services or known someone who had still demonstrated a good perception of the value of SLPT for patients needing SLPT. Of note, the familiarity of the nurses with the role of the SLP in the hospital setting was significantly associated with their perceptions of the value of SLPT. This finding could mean that the nurses have good perceptions of the value of SLPT even though the majority of them were unfamiliar with the role of the SLP in the hospital setting. The findings of the current study indicate a good awareness of the scope of practice of the SLP; however, further investigations are necessary. In an international context, this finding contrasts with another study where 49% of the nurses reported that less than 50% of nonspeaking patients were routinely referred to the SLP [20]. Likewise, a study in India reported that healthcare providers, including nurses, frequently did not refer patients with speech impairments to the SLP, showing a lack of awareness regarding the scope of practice of SLPT [25]. Although this finding merits further investigation, it is comparable to a recent study in Saudi Arabia where only 4% of the participants (N = 174 nurses) were aware of the role of the SLP in the management of dysphagia for patients with cerebral palsy, stroke, and traumatic brain injuries from traffic accidents [23].

Lastly, the other work-related variables were not found to have significant differences and associations with the nurses’ perceptions of the value of SLPT or with their understanding of the scope of practice of the SLP. Despite the fact that no significant differences and associations were evident in these aspects, nurses must be equipped with sufficient knowledge and the necessary awareness to refer a patient requiring the SLPT services to the SLP. Hence, to improve patient outcomes and ensure the delivery of quality and patient-centered care, it is important to establish better interprofessional collaboration between nurses and the SLP [1,24].

## 5. Conclusions 

The present study revealed that nurses had an overall good understanding of the value and scope of practice of the SLP. This study highlighted nurses’ understanding of the value and scope of practice of the SLP, demonstrating the need for interdisciplinary collaboration between both healthcare professions to achieve better patient outcomes in the study setting. For effective interprofessional collaboration and teamwork, it is vital for nurses to understand the value, role, and scope of practice of the SLP.

### Strengths and Limitations

The strength of this study is that it is the first to explore the knowledge and perception of registered nurses regarding the scope of practice of SLP in Saudi Arabia. However, all registered nurses were from the same medical facility which could limit the generalizability of the findings. Consequently, our interpretation of the findings should be viewed as the basis for further research.

## Figures and Tables

**Table 1 ijerph-18-10534-t001:** Work-related Characteristics of the Participants (N = 294).

Variable	Mean (SD)	Range
Years of experience in the hospital	7.65 (5.53)	0–37
	n	%
Unit		
Intensive Care Units (Adult and Neonatal)	37	12.6
Obstetric Department	26	8.8
Cardiology and Cathlab	40	13.6
Psychiatric Department	13	4.4
Emergency and Out-Patient Department	50	17.0
Pediatric Department	25	8.5
Oncology Department	32	10.9
Medical Department	40	13.6
Surgical Department	31	10.5
Shift		
Day	227	77.2
Night	67	22.8
Times per month the respondents communicate with or refer to a speech-language pathologist		
0 times	240	81.6
1–10 times	41	13.9
More than 10 times	13	4.4
Number of patients the respondents cared for with the speech-language pathologist each month		
0 times	222	75.5
1–10 times	48	16.3
More than 10 times	24	8.2
Familiarity with the role of the speech-language pathologist in an acute care hospital setting		
Not familiar	184	62.6
Somewhat familiar	77	26.2
Familiar	29	9.9
Very familiar	4	1.4
Have you or anyone you know personally received speech therapy services?		
No	256	87.1
Yes	38	12.9

**Table 2 ijerph-18-10534-t002:** Participants’ Perception of the Value of Speech-Language Pathology (N = 294).

Item	Not at All Important	Possibly Important	Important	Extremely Important	Mean (SD)
How important is a speech-language pathologist for providing services to:	n (%)	n (%)	n (%)	n (%)	
A patient who has had a stroke or brain injury	11 (3.7)	12 (4.1)	66 (22.4)	205 (69.7)	3.58 (0.74)
A patient who was on a ventilator for several days	17 (5.8)	56 (19.0)	85 (28.9)	136 (46.3)	3.16 (0.93)
A patient who is in a persistent vegetative state with tube feeding ^a^	105 (35.7)	81 (27.6)	48 (16.3)	60 (20.4)	2.21 (1.14)
A patient who is having trouble swallowing food	16 (5.4)	36 (12.2)	106 (36.1)	136 (46.3)	3.23 (0.87)
A patient who has acute vomiting due to a bowel obstruction ^a^	72 (24.5)	82 (27.9)	55 (18.7)	85 (28.9)	2.52 (1.15)
A patient who is receiving head/neck radiation	38 (12.9)	66 (22.4)	99 (33.7)	91 (31.0)	2.83 (1.01)
	Yes	No	Uncertain		
	n (%)	n (%)	n (%)		
Do you think a patient’s communicative abilities can change as a result of speech therapy?	256 (87.1)	7 (2.4)	31 (10.5)		
Do you think a patient’s swallowing abilities can change as a result of speech therapy?	200 (68.0)	20 (6.8)	74 (25.2)		
Do speech therapy services make a substantial contribution to patient care within the hospital setting?	251 (85.4)	5 (1.7)	38 (12.9)		

Note. ^a^ Reverse coded.

**Table 3 ijerph-18-10534-t003:** Results of the Tests for Differences and Associations between the Work-Related Characteristics and Perceptions of the Value of Speech-Language Pathology (N = 294).

Variable	Mean	SD	Test	*p*
Unit				
Intensive Care Units (Adult and Neonatal)	2.94	0.35	*F* = 1.99	0.057
Obstetric Department	2.83	0.46		
Cardiology and Cathlab	2.84	0.35		
Psychiatric Department	2.96	0.39		
Emergency and Out-Patient Department	2.89	0.24		
Pediatric Department	2.91	0.30		
Oncology Department	3.05	0.25		
Medical Department	3.04	0.38		
Surgical Department	2.85	0.36		
Shift				
Day	2.92	0.34	*t* = 0.31	0.760
Night	2.91	0.37		
Times per month the respondents communicate with or refer to a speech-language pathologist				
0 times	2.91	0.33	*F* = 2.10	0.125
1–10 times	3.01	0.40		
More than 10 times	2.82	0.34		
Number of patients the respondents cared for with the speech-language pathologist each month				
0 times	2.91	0.34	*F* = 2.75	0.065
1–10 times	3.01	0.34		
More than 10 times	2.82	0.34		
Have you or anyone you know personally received speech therapy services?				
No	2.90	0.35	*t =* −2.45	0.015 *
Yes	3.05	0.27		
Years of experience in the hospital			*r* = 0.03	0.565
Familiarity with the role of the speech-language pathologist in the acute care hospital setting			*r* = 0.13	0.032 *

Note. * Significant at 0.05 level.

**Table 4 ijerph-18-10534-t004:** Participants’ Understanding of the Speech-Language Pathologist’s Scope of Practice based on Scenarios (N = 294).

Scenario	Correct		Incorrect	
	n	%	n	%
Scenario 1	217	73.8	77	26.2
Scenario 2	170	57.8	124	42.2
Scenario 3	191	65.0	103	35.0
Scenario 4	247	84.0	47	16.0
Scenario 5	133	45.2	161	54.8
Scenario 6	241	82.0	53	18.0
Scenario 7	241	82.0	53	18.0
Scenario 8	123	41.8	171	58.2
Scenario 9	239	81.3	55	18.7
Scenario 10	54	18.4	240	81.6
Scenario 11	228	77.6	66	22.4
Overall		64.4%		35.6%

**Table 5 ijerph-18-10534-t005:** Results of the Tests for Differences and Associations between the Work-Related Characteristics and Understanding of the Speech-Language Pathologist’s Scope of Practice (N = 294).

Variable	Mean	SD	Test	*p*
Unit				
Intensive Care Units (Adult and Neonatal)	7.03	1.32	*F* = 2.42	0.055
Obstetric Department	6.54	1.86		
Cardiology and Cathlab	6.72	1.57		
Psychiatric Department	6.15	1.82		
Emergency and Out-Patient Department	7.64	1.41		
Pediatric Department	7.56	1.53		
Oncology Department	7.50	1.95		
Medical Department	6.95	1.57		
Surgical Department	6.97	1.94		
Shift				
Day	7.15	1.63	*t* = 1.24	0.215
Night	6.86	1.80		
Times per month the respondents communicate with or refer to a speech-language pathologist				
0 times	7.17	1.60	*F* = 1.65	0.195
1–10 times	6.66	1.89		
More than 10 times	7.00	2.12		
Number of patients the respondents cared for with the speech-language pathologist each month				
0 times	7.07	1.61	*F* = 0.15	0.861
1–10 times	7.21	1.57		
More than 10 times	7.04	2.35		
Have you or anyone you know personally received speech therapy services?				
No	7.10	1.65	*t* = 0.25	0.806
Yes	7.03	1.81		
Years of experience in the hospital			*r* = 0.05	0.360
Familiarity with the role of the speech-language pathologist in the acute care hospital setting			*r* = 0.06	0.292

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
