# Peer review of "Knowledge and Perception of Registered Nurses Regarding the Scope of Practice of Speech-Language Pathologists"

_ijerph, 2021, doi:10.3390/ijerph181910534_

Round 1
Reviewer 1 Report
A very interesting study, but the introduction and discussion need improvement. First of all, a very modest review of the available literature. The authors of the work should make every effort to properly review the literature from the last 5 years.
Author Response
A very interesting study, but the introduction and discussion need improvement. First of all, a very modest review of the available literature. The authors of the work should make every effort to properly review the literature from the last 5 years.
RESPONSE: Thank you very much for this comment. Recent literature from the last 5 years was added in the Introduction section. Correspondingly, relevant citations were also added in the Reference section and updated the order of referencing. For the Discussion section, we revised and improved it as suggested and in consideration with the valuable comments of the other reviewer.

Reviewer 2 Report
Knowledge and Perception of Registered Nurses regarding the Scope of Practice of Speech-Language Pathologists
The paper addresses some issues related to the knowledge and perception of registered nurses regarding the scope of practice of speech-language pathologists (SLP). Sample of 294 registered nurses were surveyed utilizing the Knowledge and Perception of Speech-Language Pathologists questionnaire. The analysis included descriptive statistics, tests for differences, and regression analyses. The results revealed that nursing respondents had an overall good understanding of the value and scope of practice of the SLP. Nevertheless, they frequently and wrongly responded to scenarios concerning patients who suffered from Alzheimer’s dementia, laryngeal cancer, traumatic brain injury, and stroke. The findings are useful for nurses to understand the value, role, and scope of practice of the SLP and additional educational nursing interventions and training are needed for interprofessional effectiveness.
Even though the results are not impressive the paper is interesting and comprehensible to the reader.
I have some methodological observation which need to be addressed.
-Please provide an account about validity and reliability of the instrument used,
-Moreover, in the abstract it is mentioned that linear multiple regression was used. Howeve on the Tables I cannot recognize the result of a regression analyses formally presented (including betas, esds, t-tests…) Is this Table 3? It present various test (e.g. Pearson correlation). It is not clear what the Table presents.
Author Response
Even though the results are not impressive the paper is interesting and comprehensible to the reader.
RESPONSE: Thank you very much for the positive review.
I have some methodological observation which need to be addressed.
-Please provide an account about validity and reliability of the instrument used,
RESPONSE: Thank you very much for this comment. We added this information, “The tool had been used in previous studies (12,17), and was pilot-tested among 50 nurses in Saudi Arabia before the start of the study, which yielded alpha values > 0.70 for the two subscales,” on lines 97-98.
Moreover, in the abstract it is mentioned that linear multiple regression was used. Howeve on the Tables I cannot recognize the result of a regression analyses formally presented (including betas, esds, t-tests…) Is this Table 3? It present various test (e.g. Pearson correlation). It is not clear what the Table presents.
RESPONSE: Thank you very much for this comment. For the “linear multiple regression” in the Abstract, we sincerely apologize for this typo/entry error because we did not perform such test in our paper. We deleted such error in the abstract on lines 15-16, and in the methods section on lines 139-140.

Reviewer 3 Report
The authors present a descriptive study of the knowledge and perception of registered nurses regarding the scope of practice of speech-language pathologists. A convenience sample of 294 registered nurses from a tertiary university medical center in Saudi Arabia completed the Knowledge and Perception of Speech-Language Pathologists Questionnaire (response rate was 94.4%. Analyses consist primarily of descriptive results plus a few group comparisons. The results suggest that the respondents had an overall good understanding of the value and scope of practice of the speech-language pathologist. However, the percentage of correct responses did not meet the 80% correct threshold of acceptability for certain scenarios concerning patients who suffered from Alzheimer’s dementia, laryngeal cancer traumatic brain injury, and stroke. The authors conclude that further educational interventions and training are needed for effective interprofessional collaboration and teamwork.
This is a straightforward preliminary descriptive study on a topic that has not been well studied in Saudi Arabia. Some suggestions for further strengthening the manuscript are listed below.
- Pg. 2 and pg. 3. Consider moving the information about the response rate of participants on pg. 3, lines 107-8, to the Methods section on pg. 2, Section 2.1.
- Pg. 2, lines 80-81. It would help if the authors would describe the content of the “three remaining questions”. At present, the reader needs to refer to Table 2 to figure it out.
- Methods section. It would help if the authors provided more details on the range of values associated with the variable referred to as “understanding of the speech-language pathologists’ scope of practice”, to help the reader understand and interpret the mean values shown in Table 5. Presumably it is some type of composite variable, but there is no information on how this composite variable was constructed.
- Results section, pg. 4, lines 124-135. It would help if the authors considered taking the percentages from Table 2 for the “important” and “extremely important” response categories and adding them together in the text. Doing so would help to highlight the pattern of results that the authors want to present regarding perceptions of the importance of providing services to patients with certain conditions.
- Results section, pg. 4, lines 136-142. It would help if the authors considered taking the percentages shown in Table 2 for the “not at all important” and “possibly important” categories and adding them together in the text. As mentioned above, doing so helps to enhance the presentation of the results by highlighting the pattern that the authors want the reader to see.
Author Response
This is a straightforward preliminary descriptive study on a topic that has not been well studied in Saudi Arabia.
RESPONSE: Thank you very much for your positive review.
Some suggestions for further strengthening the manuscript are listed below.
Pg. 2 and pg. 3. Consider moving the information about the response rate of participants on pg. 3, lines 107-8, to the Methods section on pg. 2, Section 2.1.
RESPONSE: Thank you very much for this comment. We moved the information on lines 107-108 to lines 86-88 on page 2, Section 2.1.
Pg. 2, lines 80-81. It would help if the authors would describe the content of the “three remaining questions”. At present, the reader needs to refer to Table 2 to figure it out.
RESPONSE: Thank you very much for this comment. We revised lines 80-81 including 78-79 because the statements were misleading which the meaning was changed during English editing. Further, the three remaining questions on this section are merely matters of opinion. There is no right or wrong answer. If the participant responds with "yes" on each question, he/she has a positive perception of SLPs, whereas if he/she responds with "no" on the questions, he/she has a negative perception of SLPs. Please revisions on lines 109-116.
Methods section. It would help if the authors provided more details on the range of values associated with the variable referred to as “understanding of the speech-language pathologists’ scope of practice”, to help the reader understand and interpret the mean values shown in Table 5. Presumably it is some type of composite variable, but there is no information on how this composite variable was constructed.
RESPONSE: Thank you very much for this comment. To receive credit for a correct response, participants must mark the SLP as one of the professionals who should be involved in the patient's treatment for each of the following items: 1, 2, 4, 5, 6, 8, 9, and 11. The SLP should not be marked for the following items: 3, 7, and 10 (these items are included as foils). It does not matter if other disciplines were/were not marked on this portion of the questionnaire; the only discipline that determines whether the response is correct/incorrect is the SLP. Please see revisions on lines 121-126.
Results section, pg. 4, lines 124-135. It would help if the authors considered taking the percentages from Table 2 for the “important” and “extremely important” response categories and adding them together in the text. Doing so would help to highlight the pattern of results that the authors want to present regarding perceptions of the importance of providing services to patients with certain conditions.
RESPONSE: Thank you very much for this comment. We revised these parts in the Results section as suggested. Please see lines 169-176.
Results section, pg. 4, lines 136-142. It would help if the authors considered taking the percentages shown in Table 2 for the “not at all important” and “possibly important” categories and adding them together in the text. As mentioned above, doing so helps to enhance the presentation of the results by highlighting the pattern that the authors want the reader to see.
RESPONSE: Thank you very much for this comment. We revised these parts in the Results section. By doing so, there was a notable change of presenting the result of one question regarding a patient suffering from acute vomiting due to a bowel obstruction (52.4%) that was appropriately perceived by the respondents by rating it as not all important or possibly important. This finding shows that the nurses had a good perception of the value of the SLPT in this situation. The previous presentation was opposite. Please see lines 182-185. Correspondingly, the discussion part related to this change was also revised. Please see lines 252-257.

Round 2
Reviewer 1 Report
Thank you very much for improving the manuscript.